# Universally Expressive Communication in Multi-Agent Reinforcement Learning

**Matthew Morris**
InstaDeep Ltd. & University of Oxford
`matthew.morris@cs.ox.ac.uk`

**Thomas D. Barrett**
InstaDeep Ltd.
`t.barrett@instadeep.com`

**Arnu Pretorius**
InstaDeep Ltd.
`a.pretorius@instadeep.com`

## Abstract

Allowing agents to share information through communication is crucial for solving complex tasks in multi-agent reinforcement learning. In this work, we consider the question of whether a given communication protocol can express an arbitrary policy. By observing that many existing protocols can be viewed as instances of graph neural networks (GNNs), we demonstrate the equivalence of joint action selection to node labelling. With standard GNN approaches provably limited in their expressive capacity, we draw from existing GNN literature and consider augmenting agent observations with: (1) unique agent IDs and (2) random noise. We provide a theoretical analysis as to how these approaches yield universally expressive communication, and also prove them capable of targeting arbitrary sets of actions for identical agents. Empirically, these augmentations are found to improve performance on tasks where expressive communication is required, whilst, in general, the optimal communication protocol is found to be task-dependent.

## 1 Introduction

Communication lies at the heart of many multi-agent reinforcement learning (MARL) systems. In MARL, multiple agents must account for each other's actions during both training and execution and, indeed, solving complex tasks in high-dimensional spaces often requires a cooperative joint policy that is difficult, or even impossible, to learn independently. Therefore, allowing agents to share information is crucial and how best to achieve this has remained a keen area of research since the seminal proposals of learned communication by Foerster et al. [14] and Sukhbaatar et al. [51]. Whilst no single universally-adopted approach has emerged, considerations for MARL communication include inductive biases that aid learning. For example, an agent's policy should often not depend on the order in which messages are received at a given time step. i.e. be *permutation invariant*.

In this context, graph neural networks (GNNs) provide a rich framework for MARL communication. It is natural to consider agents as nodes in a graph, with communication channels corresponding to edges between them. GNNs are specifically designed to respect this (typically non-Euclidian) structure [5] and, indeed, many of the most successful MARL communication models fall within this paradigm, including CommNet [51], IC3Net [49], GA-Comm [29], MAGIC [37], Agent-Entity Graph [2], IP [44], TARMAC [9], IMMAC [52], DGN [24], VBC [64], MAGNet [33], and TMC [65]. Other models such as ATOC [23] and BiCNet [43] do not fall within the paradigm since they use LSTMs for combining messages, which are not permutation invariant, and models such as RIAL, DIAL [14], ETCNet [21], and SchedNet [25] do not since they used a fixed message-passing structure.

36th Conference on Neural Information Processing Systems (NeurIPS 2022).

Figure 1: A pair of graphs indistinguishable by 1-WL

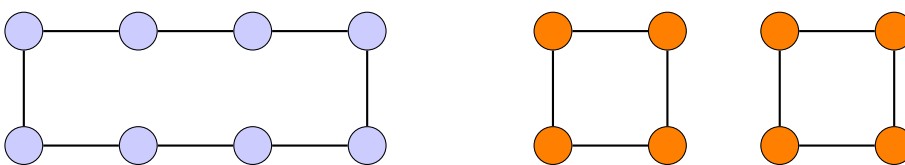

However, although traditional GNNs – such as those used in MARL to date – can readily provide permutation invariant communication, they are not universally expressive.

The expressivity of GNNs is often considered in the context of the 1-WL graph coloring algorithm [60]. In brief, 1-WL tests if two graphs are non-isomorphic by iteratively re-coloring the nodes and has been proven to *not* be universally expressive (i.e. there exist non-isomorphic graphs that 1-WL can't distinguish). Moreover, Morris et al. [34] and Xu et al. [61] proved that for any two non-isomorphic graphs indistinguishable by 1-WL, there is no GNN that can produce different outputs for those two graphs. An example of such graphs is given in Figure 1. This direct correspondence between GNNs and 1-WL equivalently limits the expressivity of any MARL communication built on top of GNNs. Whist higher-order GNN architectures which go beyond 1-WL expressivity have been proposed (see [35] for an overview), many of these models do not scale well and are computationally infeasible in practice. However, recent works have shown that augmenting the node features can provide an alternative path to increased expressivity without computationally expensive architectural changes [1, 10]. It is then natural to ask if, and how, these advancements can be brought into the MARL setting.

In this paper, we investigate the effectiveness of GNNs for universally expressive communication in MARL. We define *Graph Decision Networks* (GDNs), a framework for MARL communication which captures many of the most successful methods. We highlight the correspondence between GDNs and the node labeling problem in GNNs, thus making concrete the limits of GDN expressivity. For moving beyond these limits, we consider two augmentations from the GNN literature – random node initialization (RNI) [1] and colored local iterative procedure (CLIP) [10], where random noise and unique labels are added to graph nodes, respectively. We provide a theoretical analysis as to how these algorithms yield universally expressive communication in MARL, and also prove their ability to solve coordination problems where the optimal policy requires arbitrary sets of actions from identical agents. We then perform an empirical study where we augment several state-of-the-art MARL communication algorithms with RNI and unique labels. By evaluating performance across both standard benchmarks and specifically designed tasks, we show that when complex non-local coordination or symmetry breaking is required, universally expressive communication can provide significant performance improvements. However, in more moderate cases, augmented communication can reduce convergence speeds and result in suboptimal policies. Therefore, whilst more expressive GNN architectures are required to improve performance on certain problems, a more complete picture relating expressivity to downstream performance remains an open question for future work.

## 2 Background

**Multi-Agent Reinforcement Learning**   We consider the setting of Decentralized Partially Observable Markov Decision Processes [39] augmented with communication between agents. At each timestep $t$ every agent $i \in \{1, ..., N\}$ gets a local observation $o_i^t$, takes an action $a_i^t$, and receives a reward $r_i^t$. We consider two agent paradigms: value-based [53] and actor-critic [15, 32]. For brevity, we collectively refer to the policy network in actor-critic methods and the Q-network in value-based methods as the *actor network*. In this paper, we consider by default parameter sharing between agent's networks, which is often used to yield faster and more stable training in MARL [14, 18, 45, 62].

**Graph Neural Networks**   GNNs can refer to a large variety of models; in this paper, we define the term to correspond to the definition of Message Passing Neural Networks (MPNNs) by Gilmer et al. [17], which is the most common GNN architecture. Notable instances of this architecture include Graph Convolutional Networks (GCNs) [12], GraphSAGE [19], and Graph Attention Networks (GATs) [56]. A GNN consists of multiple message-passing layers, each of which updates the node

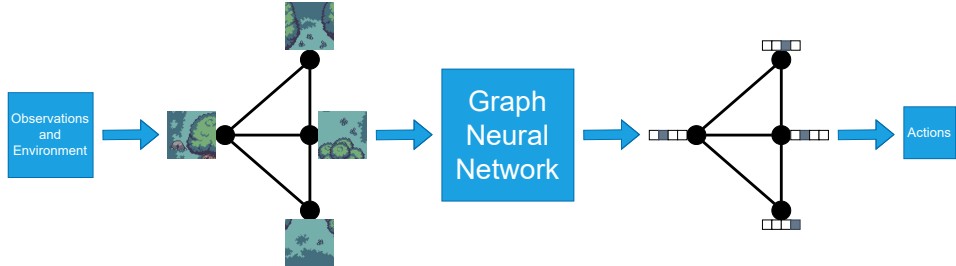

Figure 2: The Graph Decision Network (GDN) framework

attributes / labels (terms used interchangeably). For layer $m$ and node $i$ with current attribute $v_i^m$, the new attribute $v_i^{m+1}$ is computed as

$$v_i^{m+1} := f_{\text{update}}^{\theta_m}(v_i^m,\ f_{\text{aggr}}^{\theta_m'}(\{v_j^m \mid j \in N(i)\}),\ f_{\text{read}}^{\theta_m''}(\{v_j^m \mid j \in V(G)\}))$$

where $N(i)$ is all nodes with edges connecting to $i$ and $\theta_m, \theta_m', \theta_m''$ are the (possibly trainable) parameters of the update, aggregation, and readout functions for layer $m$. Parameters may be shared between layers, e.g. $\theta_0 = \theta_1$. The functions $f_{\text{aggr}}^{\theta_m'}, f_{\text{read}}^{\theta_m''}$ are permutation invariant. Importantly, GNNs are invariant / equivariant graph functions.

## 3 Expressivity of Multi-Agent Communication

### 3.1 Graph Decision Networks

Many of the most successful MARL communication methods can be captured within the following framework. At each time step, define an attributed graph $G = (V, E)$ with nodes $V(G) := \{\text{all agents}\}$, edges $E(G) := \{(i, j) \mid \text{agent } i \text{ is communicating with } j\}$, and for all agents $i$, the node $i$ is labeled with the observation of $i$. This graph is passed through a GNN $f$ which outputs values for each node and passes each resulting node value through the actor network of the corresponding agent. Assuming that the actor networks use shared weights (i.e. the same neural network is used for each actor), we can substitute them for a final GNN layer $M$, where $f_{\text{update}}^{\theta_M}(v, \sim) := P(v)$ and $P$ represents the shared actor network. We refer to communication methods that fall within this paradigm as *graph decision networks* (GDNs). The framework is illustrated visually in Figure 2.

Given the above assumption of shared weights, any GDN simply reduces to a GNN node labelling problem, where the correct label for a given node is the corresponding actor network output that collectively maximizes the joint reward (or the individual reward, depending on how the agent is trained). Going forward, we only deal with such GDNs (ones with a shared actor network). In the case of stochastic policies [15, 63], the target labels are parameters of output distributions, instead of atomic actions. This applies to both discrete and continuous distributions. Note that this is not how RL agents are actually trained (i.e. they use reward signals, not supervised learning).

However, given that this paper aims to analyze expressivity, we argue that it does not matter how the GDN is trained. All that expressivity is concerned with is the *ability* of a model to produce a certain output, not how the training paradigm causes the model to converge to the solution. All we have to know is that there are "optimal" actor network outputs for each agent, under some metric of optimality, and then we can reason about the ability of the model to provide these outputs. Scenarios with heterogeneous agents can still be considered within this paradigm, by allocating a portion of the observations to indicate the agent type (e.g. through a one-hot encoding) [54]. Models with recurrent networks also fall within the paradigm, where the hidden or cell states for the networks can be considered as part of the agent observations.

Due to the reduction of GDNs to a GNN node labelling problem, GDNs suffer from the same expressivity limits as GNNs, about which there is a plethora of work [4, 8, 16, 30, 31, 34, 38, 40, 61]. These are expanded upon in Appendix A.4. For our analysis, we focus particularly on ways to achieve

Figure 3: A simple example of symmetry breaking

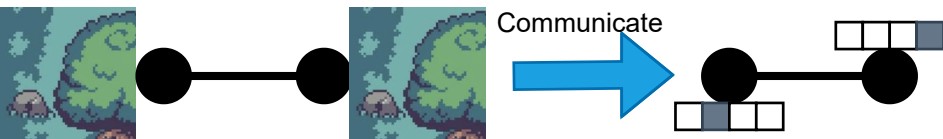

universal Weisfeiler-Lehman expressivity, but note that the above reduction unlocks many tools for reasoning about the expressivity of GDNs.

## 3.2 Desired Properties of MARL Communication

Whilst conventional GDNs cannot capture functions with expressive power beyond 1-WL [34, 61], recent GNN architectures have been proposed to achieve expressivity beyond 1-WL, even ones which are able to express any equivariant graph function. We can use these insights to construct more expressive GDNs. However, we note that classes of models which always yield equivariant functions are not necessarily desirable, since they cannot break symmetries between agents when required. Many MARL environments require agents to coordinate, needing some joint action to solve the task. However, if agents have identical observations and communication graph structure in a pure GDN framework, there is no way for them to disambiguate between each other and distribute the required actions amongst themselves. For a simple example, consider a setting where two agents have identical observations but must take opposite actions – then the only way for them to solve the environment is to communicate in such a way that they can break this symmetry and take different actions from one another. This example is illustrated in Figure 3.

More formally, since GNNs are equivariant graph functions, GDNs are equivariant functions on the agent observations and communication graph structure. This means that agents within the same *graph orbit* will always produce the same output. For a graph $G$, consider two nodes $u, v \in G$. If there exists an automorphism $\alpha$ of $G$ such that $\alpha(u) = v$, then $u$ and $v$ are said to be *similar nodes*. The relation *is similar to* forms an equivalence relation on the nodes of G. Each equivalence class is called an *orbit*. Intuitively, every node in an orbit "has the same structure". We denote the set of all orbits of $G$ by $R(G)$: this forms a partition of $V(G)$.

**Theorem 1.** *Given a GDN $f$, observations $O = \{o_1, ..., o_n\}$, and communication graph $G$ such that nodes $i$ and $j$ are similar in $G$ and $o_i = o_j$, then it holds that $f(O)_i = f(O)_j$.*

Full proofs for all theorems in this paper can be found in Appendix B. We formally state the desired behaviour of GDNs – which we refer to as *symmetry breaking* – that would enable them to solve such coordination problems. Given a graph $G$ with orbits $R(G)$, a GDN $g$ ought to be able to produce, or *target*, a multiset of labels $A_k$ for each orbit $r_k$:

$$\forall r_k \in R(G), \{g(G)_i \mid i \in r_k\} = A_k,$$

where $g(G)_i$ is the output of $g(G)$ for agent $i$. Thus, ideally, MARL communication methods should possess all the following properties: (1) universal expressivity for equivariant graph functions, (2) symmetry breaking for coordination problems, and (3) computational efficiency. We apply two existing GNN augmentations to GDNs to achieve this, both of which come with minimal extra computational cost. In the following section, we provide theorems which prove that the first two properties are satisfied by these augmentations.

## 3.3 Expressive Graph Decision Networks

**Random Node Initialization** Sato et al. [47] propose augmenting GNNs with *random node initialization* (RNI), where for each node in the input graph, a number of randomly sampled values are concatenated to the original node attribute. For all graphs / nodes, the random values are sampled from the same distribution. Abboud et al. [1] prove that such GNNs are universal and can approximate any permutation invariant / equivariant graph function. Technically, random initialization breaks the node invariance in GNNs, since the result of the message passing will depend on the structure of the

graph as well as the values of the random initializations. However, when one views the model as computing a random variable, the random variable is still invariant when using RNI. In expectation, the mean of random features will be used for GNN predictions, and is the same across each node. However, the variability of the random samples allows the GNN to discriminate between nodes that have different random initializations, breaking the 1-WL upper bound.

Abboud et al. [1] formally state and prove a universal approximation result for invariant graph functions. They note that it can be extended to equivariant functions, which is what GDNs are. As such, we adapt and state the theorem for equivariant functions. Let $G_n$ be the class of all $n$-node graphs. Let $f : G_n \to \mathbb{R}^n$, a graph function which outputs a real value for each node in $V(G)$. We say that a randomized function $X$ that associates with every graph $G \in G_n$ a sequence of random variables $X_1(G), X_2(G), ..., X_n(G)$, one for each node, is an $(\epsilon, \delta)$-approximation of $f$ if for all $G \in G_n$ it holds that $\forall i \in \{1, 2, ..., n\}, \Pr(|f(G)_i - X_i(G)| \leq \epsilon) \geq 1 - \delta$, where $f(G)_i$ is the output of $f(G)$ for node $i$. Note that a GNN $h$ with RNI computes such functions $X$. If $X$ is computed by $h$, we say that $h$ $(\epsilon, \delta)$-approximates $f$. We can now state the following theorem:

**Theorem 2.** *Let $n \geq 1$ and let $f : G_n \to \mathbb{R}^n$ be equivariant. Then for all $\epsilon, \delta > 0$, there is a GNN with RNI that $(\epsilon, \delta)$-approximates $f$.*

Such GNNs are also able to solve symmetry-breaking coordination problems by using RNI to disambiguate between otherwise identical agents. To formally state this property, we need to define $(\epsilon, \delta)$-approximation for sets. We say that two multisets $A, B$ containing random variables are $(\epsilon, \delta)$-equal, denoted $A \cong_{\epsilon, \delta} B$, if there exists a bijection $\tau : A \to B$ such that $\forall a \in A, \Pr(|a - \tau(a)| \leq \epsilon) \geq 1 - \delta$.

**Theorem 3.** *Let $n \geq 1$ and consider a set $T$, where each $(G, A) \in T$ is a graph-labels pair, such that $G \in G_n$ and there is a multiset of target labels $A_k \in A$ for each orbit $r_k \in R(G)$, with $|A_k| = |r_k|$. Then for all $\epsilon, \delta > 0$ there is a GNN with RNI $g$ which satisfies:*

$$\forall (G, A) \in T \ \forall r_k \in R(G), \{g(G)_i \mid i \in r_k\} \cong_{\epsilon, \delta} A_k$$

In the GDN case, adding RNI means concatenating noise to the agent observations, thus achieving universal approximation and enabling the solving of symmetry-breaking coordination problems.

**Unique Node Identifiers** Dasoulas et al. [10] augment GNNs with a coloring scheme to define *colored local iterative procedure* (CLIP). They use colors to differentiate otherwise identical node attributes, with $k$-CLIP corresponding to $k$ different colorings being sampled and maximized over. They prove theoretically that when maximizing over all such possible colorings, $\infty$-CLIP can represent any invariant graph function.

Assigning nodes unique IDs is equivalent to 1-CLIP, since this guarantees that every node with identical attributes will have a unique "color": its particular unique ID. Therefore, we can leverage the universality result for 1-CLIP (Theorem 4 in [10]), which states that with any given degree of precision, 1-CLIP can approximate any invariant graph function. However, Dasoulas et al. [10] note that such solutions may be difficult to converge to and require a large number of training steps in practice. Intuitively, this is because the GNN has to learn to deal with $n!$ permutations of unique IDs. Similarly to the RNI case, we extend their theorem to equivariant functions.

**Theorem 4.** *Let $n \geq 1$ and let $f : G_n \to \mathbb{R}^n$ be equivariant. Then for all $\epsilon > 0$, there is a GNN with unique node IDs that $\epsilon$-approximates $f$.*

Such GNNs can also solve symmetry-breaking coordination problems, in a similar way to ones with RNI. We say that two multisets $A, B$, which do not contain random variables, are $\epsilon$-equal, denoted $A \cong_{\epsilon} B$, if there exists a bijection $\tau : A \to B$ such that $\forall a \in A, |a - \tau(a)| \leq \epsilon$.

**Theorem 5.** *Let $n \geq 1$ and consider a set $T$, where each $(G, A) \in T$ is a graph-labels pair, such that $G \in G_n$ and there is a multiset of target labels $A_k \in A$ for each orbit $r_k \in R(G)$, with $|A_k| = |r_k|$. Then for all $\epsilon > 0$ there is a GNN with unique node IDs $g$ which satisfies:*

$$\forall (G, A) \in T \ \forall r_k \in R(G), \{g(G)_i \mid i \in r_k\} \cong_{\epsilon} A_k$$

In the GDN case, this means that by giving each agent a unique ID in its observations, we can achieve universal approximation and enabling the solving of symmetry-breaking coordination problems.

Table 1: Architecture of the Baselines

| Name | Communication Graph | Agents | GNN Architecture |
|------|---------------------|--------|------------------|
| CommNet [51] | Complete (or environment-based) | Recurrent A2C | Sum Aggregation |
| IC3Net [49] | Complete + Gating | Recurrent A2C | Sum Aggregation |
| TarMAC [9] | Complete + Learned Soft Edges | Recurrent A2C | GAT |
| T-IC3Net [49, 9] | Gating + Learned Soft Edges | Recurrent A2C | GAT |
| MAGIC [37] | Learned | Recurrent A2C | GAT |
| DGN [24] | Environment-based | Q-network | GCN |

## 4 Experiments

### 4.1 Methods

**Baselines**  For evaluation, we adopt a diverse selection of MARL communication methods which fall under the GDN paradigm. These are shown in Table 1, along with the communication graph structure, agent model, and GNN architecture. We use the code provided by Jiang et al. [24], Niu et al. [37] as starting points. All of the implementations are extended to support multiple rounds of message-passing and the baselines are augmented with the ability for their communication to be masked by the environment (e.g. based on distance or obstacles in the environment). We fix the number of message-passing rounds to be 4 and otherwise use the original models and hyperparameters from [24, 37]. Full experiment and hyperparameter details can be found in Appendix C, and full results are shown in Appendix D.

**Environments**  **Predator-Prey** [9, 27, 29, 37, 49] and **Traffic Junction** [9, 27, 29, 37, 49, 51] are common MARL communication benchmarks. In Predator-Prey, predator agents are tasked with capturing prey and in Traffic Junction, agents need to successfully navigate a traffic intersection (full descriptions of each environment are given in Appendix C.2). We perform evaluations on these benchmarks to test how well our universally expressive GDN models perform when there is not necessarily a benefit to having communication expressivity beyond 1-WL. We also introduce two new environments, Drone Scatter and Box Pushing, to respectively test symmetry-breaking and communication expressivity beyond 1-WL.

**Drone Scatter**  consists of 4 drones in a homogeneous field surrounded by a fence. Their goal is to move around and find a target hidden in the field, which they can only notice when they get close to. The drones do not have GPS and can only see directly beneath them using their cameras, as well as observing their last action. The best way for them to locate the target is to split up and search in different portions of the field, despite them all having the same observations; thus, they are given rewards for splitting up.

**Box Pushing**  consists of 10 robots in a 12x12 construction site, which has boxes within that need to be moved to the edge of the site: the clearing area. Robots attach themselves to boxes before they can move them; when attached, robots can no longer see around themselves. Free-roaming robots can communicate with any other free-roaming robots, but attached robots can only communicate with the robots directly adjacent to them. The environment either spawns with one large box or two small boxes and agents spawn already attached. 4 attached robots all moving in the same direction are needed to move a small box, and 8 all power moving in the same direction to move a large box. To solve the environment, the robots need to be able to communicate with each other to figure out which type of box they are on and all push correctly, at the same time, and in the same direction. Since the communication graphs corresponding to the scenarios with small and large boxes are 1-WL indistinguishable, communication beyond 1-WL is needed to optimally solve the environment.

**Evaluation Procedure**  We augment baseline communication methods with RNI and unique IDs to perform our evaluations. Agent IDs are represented by one-hot encodings and "0.25 RNI" refers to 25% of the observation space being randomly initialized. We sample each RNI value uniformly from $[-1, 1]$. For each scenario and for every baseline communication method, we compare 4 models: the baseline without modifications, the baseline augmented with unique IDs for each agent, the baseline augmented with 0.75 RNI, and finally 0.25 RNI. The only exceptions are the Drone Scatter evaluations, where 0.25 RNI is not used since the observation space is not large enough, and the

Table 2: Mean and 95% confidence interval for Easy Traffic Junction across all baselines

| Baseline | Metric | Baseline | Unique IDs | 0.75 RNI | 0.25 RNI |
|---|---|---|---|---|---|
| CommNet | Success | $1 \pm 0$ | $1 \pm 0$ | $1 \pm 0$ | $1 \pm 0$ |
| DGN | Success | $0.987 \pm 0$ | $0.99 \pm 0$ | $0.848 \pm 0.15$ | $\mathbf{0.996 \pm 0}$ |
| IC3Net | Success | $1 \pm 0$ | $1 \pm 0$ | $1 \pm 0$ | $0.986 \pm 0.02$ |
| MAGIC | Success | $0.634 \pm 0.11$ | $0.764 \pm 0.13$ | $0.684 \pm 0.11$ | $\mathbf{0.787 \pm 0.09}$ |
| TarMAC | Success | $0.994 \pm 0.01$ | $1 \pm 0$ | $0.933 \pm 0.04$ | $1 \pm 0$ |
| T-IC3Net | Success | $1 \pm 0$ | $0.998 \pm 0$ | $0.94 \pm 0.04$ | $0.974 \pm 0.04$ |

Drone Scatter experiments using stochastic evaluation, where DGN is not used since it does not support stochastic evaluation.

For each run, corresponding to a random initialization (one seed) of the model in question, we perform periodic evaluations during training. Each epoch consists of 5000 training episodes, after which 100 evaluation episodes are used to report aggregate metric scores, yielding an evaluation score for the model after every epoch. Following an established practice in MARL evaluation [15, 22, 41, 46, 59, 63, 66], we take the value of a metric for a run to be the *best* value achieved during training, so that our metrics are robust against runs which converge at some point and then degrade in performance as they continue to train. In such cases, one would use the parameters from the best performing model found during training for real-world evaluation; thus, that performance makes more sense to report than the model performance once training has finished. We utilize 10 seeds for Box Pushing experiments and 5 for all others. For each scenario, metric, baseline communication method, and variant thereof, we report the mean metric value across all seeds and a 95% confidence interval. To calculate the confidence interval, we assume a normal distribution and compute the interval as $1.96 \times$ SEM (standard error of the mean). Finally, for all Box Pushing experiments, we make use of a form of hybrid imitation learning to help deal with exceptionally sparse rewards (full details are given in Appendix C.3).

## 4.2 Results

**Benchmark Environments** Experimental results on the benchmark environments are shown in Table 2 for Easy Traffic Junction, Table 3 for Predator-Prey, and Table 4 for Medium Traffic Junction. In general, unique IDs tends to perform comparably to the baseline. The only exception to this is for IC3Net on Medium Traffic Junction, where unique IDs struggle.

0.75 RNI is categorically the worst method, consistently getting outperformed by all other methods and only coming out on top for MAGIC on Medium Traffic Junction, which is not significant due to the instability of that set of results. 75% of observations being randomly initialized appears far too much for the system to be able to learn effective policies. However, universality results still hold for lower ratios of RNI.

0.25 RNI exhibits strong performance on Easy Traffic Junction, always solving the environment and almost always outperforming the baseline. However, on sparse-reward problems (such as Predator-Prey and, to a lesser extent, Medium Traffic Junction) RNI methods typically take longer to converge than the baseline and unique IDs, and 0.25 RNI can struggle to reach the performance of baseline methods. This aligns with Abboud et al. [1]'s observation that GNNs with RNI take significantly longer to converge than normal GNNs. This is only is exacerbated in a MARL setting with sparse reward signals, where slow convergence is expected regardless of the RNI augmentation. Indeed, on all examples, RNI methods typically take longer to converge than the baselines and unique IDs.

Overall, we conclude that both unique IDs and 0.25 RNI achieve sufficient performance on the benchmarks to qualify them for use, especially given that the extra expressivity they provide is not strictly necessary. With respect to the different baselines, we note that simple baselines such as CommNet work the best when the optimal policy is also simple, such as for Easy Traffic Junction, but that more sophisticated baselines outperform them on the complex environments. We also note the very unstable performance of MAGIC for the Traffic Junction environments.

**Weisfeiler-Lehman Expressivity** Results for the Box Pushing environment are shown in Table 5. 0.25 RNI is the clear winner, achieving the top performance across almost all baselines and only

Table 3: Mean and 95% confidence interval for Predator-Prey across all baselines

| Baseline | Metric | **Baseline** | **Unique IDs** | **0.75 RNI** | **0.25 RNI** |
|---|---|---|---|---|---|
| CommNet | Success | $0.88 \pm 0.03$ | $\mathbf{0.908 \pm 0.02}$ | $0.194 \pm 0.02$ | $0.476 \pm 0.05$ |
| DGN | Success | $0.014 \pm 0$ | $0.016 \pm 0$ | $0.026 \pm 0.03$ | $\mathbf{0.032 \pm 0.01}$ |
| IC3Net | Success | $\mathbf{0.952 \pm 0}$ | $0.93 \pm 0.02$ | $0.454 \pm 0.08$ | $0.933 \pm 0.02$ |
| MAGIC | Success | $\mathbf{0.892 \pm 0.02}$ | $0.888 \pm 0.05$ | $0.112 \pm 0.03$ | $0.451 \pm 0.09$ |
| TarMAC | Success | $0.169 \pm 0.09$ | $\mathbf{0.24 \pm 0.11}$ | $0.068 \pm 0.01$ | $0.086 \pm 0.02$ |
| T-IC3Net | Success | $\mathbf{0.938 \pm 0.02}$ | $\mathbf{0.938 \pm 0.01}$ | $0.27 \pm 0.02$ | $0.913 \pm 0.02$ |

Table 4: Mean and 95% confidence interval for Medium Traffic Junction across all baselines

| Baseline | Metric | **Baseline** | **Unique IDs** | **0.75 RNI** | **0.25 RNI** |
|---|---|---|---|---|---|
| CommNet | Success | $0.761 \pm 0.31$ | $\mathbf{0.793 \pm 0.33}$ | $0.046 \pm 0$ | $0.614 \pm 0.11$ |
| DGN | Success | $\mathbf{1 \pm 0}$ | $\mathbf{1 \pm 0}$ | $0.062 \pm 0$ | $0.619 \pm 0.4$ |
| IC3Net | Success | $\mathbf{0.971 \pm 0.04}$ | $0.804 \pm 0.1$ | $0.588 \pm 0.03$ | $0.855 \pm 0.13$ |
| MAGIC | Success | $0.551 \pm 0.28$ | $0.526 \pm 0.33$ | $\mathbf{0.734 \pm 0.21}$ | $0.4 \pm 0.35$ |
| TarMAC | Success | $\mathbf{0.064 \pm 0}$ | $0.052 \pm 0$ | $0.05 \pm 0$ | $0.054 \pm 0.01$ |
| T-IC3Net | Success | $0.89 \pm 0.17$ | $0.909 \pm 0.08$ | $0.362 \pm 0.18$ | $\mathbf{0.962 \pm 0.02}$ |

slightly worse performance in the others. It is never outperformed by the baseline. This indicates that when communication expressivity beyond 1-WL is helpful for solving the environment, using RNI is the clear choice. Across all but one method, unique IDs also outperformed the baseline, demonstrating the benefit of having higher expressivity. However, unique IDs tend to yield less stable solutions and less effective policies than 0.25 RNI. We postulate that it is easier for agents to overfit on the particular unique IDs used, since they are deterministically assigned. On the other hand, using RNI encourages the agents to learn policies which respect the permutation invariance between agents since agents will receive different random observation augmentations at each time step.

For completeness, we note that expressivity beyond 1-WL is not strictly needed to solve the Box Pushing environment, as demonstrated by several baselines achieving a "ratio cleared" score greater than 0.5, meaning they learned to sometimes clear both types of boxes. This is because the environment can be solved, albeit inefficiently, by recurrent policies which learn to alternate actions between "normal" and "power" moves each time step. Such policies are guaranteed to move the box every 2 time steps.

Figure 4: Training curves for IC3Net and CommNet on the benchmark communication environments

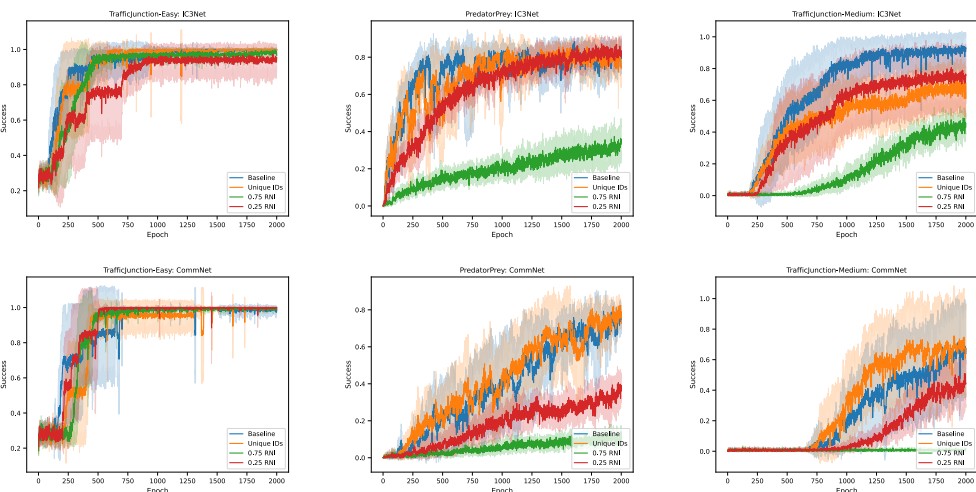

Table 5: Mean and 95% confidence interval for Box Pushing across all baselines

| Baseline | Metric | **Baseline** | **Unique IDs** | **0.75 RNI** | **0.25 RNI** |
|---|---|---|---|---|---|
| CommNet | Ratio Cleared | $0.786 \pm 0.08$ | $\mathbf{0.829 \pm 0.08}$ | $0.768 \pm 0.08$ | $0.795 \pm 0.09$ |
| DGN | Ratio Cleared | $0.603 \pm 0$ | $0.756 \pm 0.06$ | $\mathbf{0.958 \pm 0}$ | $0.957 \pm 0.01$ |
| IC3Net | Ratio Cleared | $0.49 \pm 0.15$ | $0.617 \pm 0.14$ | $0.34 \pm 0.18$ | $\mathbf{0.676 \pm 0.06}$ |
| MAGIC | Ratio Cleared | $0.958 \pm 0.04$ | $0.985 \pm 0.01$ | $0.975 \pm 0.04$ | $\mathbf{0.998 \pm 0}$ |
| TarMAC | Ratio Cleared | $0.629 \pm 0.14$ | $0.578 \pm 0.11$ | $0.662 \pm 0.06$ | $\mathbf{0.679 \pm 0.06}$ |
| T-IC3Net | Ratio Cleared | $0.558 \pm 0.13$ | $0.596 \pm 0.11$ | $0.458 \pm 0.18$ | $\mathbf{0.643 \pm 0.15}$ |

Table 6: Mean and 95% confidence interval for Drone Scatter across all baselines except DGN, including a purely random agent. Stochastic evaluation

| Baseline | Metric | **Baseline** | **Unique IDs** | **0.75 RNI** |
|---|---|---|---|---|
| CommNet | Pairwise Distance | $11.34 \pm 0.9$ | $\mathbf{12.08 \pm 1.12}$ | $8.687 \pm 1.4$ |
| | Steps Taken | $11.5 \pm 0.26$ | $\mathbf{9.767 \pm 0.32}$ | $11.74 \pm 1.39$ |
| IC3Net | Pairwise Distance | $9.108 \pm 1.45$ | $\mathbf{13.3 \pm 0.71}$ | $10.99 \pm 0.38$ |
| | Steps Taken | $11.94 \pm 0.84$ | $\mathbf{10.13 \pm 0.25}$ | $11.66 \pm 0.22$ |
| MAGIC | Pairwise Distance | $7.693 \pm 1.47$ | $\mathbf{12.59 \pm 1}$ | $7.216 \pm 0.76$ |
| | Steps Taken | $13.05 \pm 0.58$ | $\mathbf{11.12 \pm 1.05}$ | $13.54 \pm 0.21$ |
| TarMAC | Pairwise Distance | $7.448 \pm 0.89$ | $\mathbf{10.26 \pm 0.69}$ | $8.486 \pm 0.36$ |
| | Steps Taken | $13.49 \pm 0.1$ | $\mathbf{10.7 \pm 0.35}$ | $12.85 \pm 0.57$ |
| T-IC3Net | Pairwise Distance | $8.891 \pm 0.27$ | $\mathbf{12.9 \pm 0.78}$ | $9.552 \pm 0.57$ |
| | Steps Taken | $12.28 \pm 0.6$ | $\mathbf{10.33 \pm 0.46}$ | $12.22 \pm 0.82$ |
| Random | Pairwise Distance | $5.8 \pm 0.02$ | – | – |
| | Steps Taken | $17.39 \pm 0.04$ | – | – |

**Symmetry Breaking**    Results for the Drone Scatter experiments are shown in Table 6 and Table 7, for agents with stochastic and greedy evaluation respectively. Across all of them, unique IDs exhibits consistently superior performance than 0.75 RNI and the baseline, since it deterministically breaks the symmetry between agents and allows them to split up easily to solve the environment. In the stochastic case, 0.75 RNI performs similarly to the baseline, but is markedly superior to the baseline in the greedy case. The comparison to a purely random agent indicates that the models are learning much more effective policies than just moving around at random.

Baseline methods with stochastic evaluation achieve consistently higher pairwise distances than their greedy counterparts, meaning they are learning to split up to find the target. They are capable of this due to a combination of 3 things: a stochastic policy, recurrent networks, and agents observing their last actions. Initially, agents cannot differentiate between each other, and all produce the same action distribution. However, if the distribution is diverse, then they are expected to produce different actions since they sample randomly from this distribution, which are observed in the next time step. The different observations lead to different hidden states in the recurrent networks, effectively changing the observations between agents and allowing them to differentiate between each other. This is not the case for greedy action selection, which is common when doing policy evaluation in MARL, where the actions chosen will always be the same.

## 5   Conclusion

We introduce GDNs, a framework for MARL communication, and formally show how it corresponds to the node labelling problem in GNNs. Our theoretical contributions use this observation to demonstrate that existing MARL communication methods relying on conventional GNN architectures have provably limited expressivity. Driven by this, we prove how augmenting agent observations with unique IDs or random noise yields universally expressive invariant communication in MARL, whilst also providing desirable properties such as being able to perform symmetry-breaking: targeting arbitrary sets of joint actions for identical agents.

Experimentally, we compare these augmentations across 6 different MARL communication baselines that fall within the GDN paradigm, using 3 benchmark communication environments and 2 tasks

Table 7: Mean and 95% confidence interval for Drone Scatter across all baselines. Greedy evaluation

| Baseline | Metric | **Baseline** | **Unique IDs** | **0.75 RNI** |
|---|---|---|---|---|
| CommNet | Pairwise Distance | $8.849 \pm 0.63$ | $\mathbf{13.28 \pm 1.27}$ | $8.589 \pm 1.35$ |
|  | Steps Taken | $13.79 \pm 0.12$ | $\mathbf{9.554 \pm 0.33}$ | $12.62 \pm 1.19$ |
| DGN | Pairwise Distance | $3.221 \pm 0.18$ | $\mathbf{4.427 \pm 0.67}$ | $3.706 \pm 0.83$ |
|  | Steps Taken | $13.36 \pm 0.15$ | $\mathbf{13.27 \pm 0.21}$ | $13.46 \pm 0.14$ |
| IC3Net | Pairwise Distance | $7.69 \pm 1.03$ | $\mathbf{14.09 \pm 0.54}$ | $11 \pm 0.86$ |
|  | Steps Taken | $13.25 \pm 0.4$ | $\mathbf{10.14 \pm 0.2}$ | $11.42 \pm 0.48$ |
| MAGIC | Pairwise Distance | $6.61 \pm 1.28$ | $\mathbf{12.58 \pm 0.6}$ | $7.107 \pm 1.59$ |
|  | Steps Taken | $13.27 \pm 0.18$ | $\mathbf{11.84 \pm 0.68}$ | $13.61 \pm 0.25$ |
| TarMAC | Pairwise Distance | $8.666 \pm 0.28$ | $\mathbf{12.09 \pm 0.73}$ | $8.999 \pm 0.94$ |
|  | Steps Taken | $13.73 \pm 0.21$ | $\mathbf{11.01 \pm 0.86}$ | $12.19 \pm 0.82$ |
| T-IC3Net | Pairwise Distance | $7.28 \pm 0.69$ | $\mathbf{13.51 \pm 0.98}$ | $10.87 \pm 1.17$ |
|  | Steps Taken | $13.96 \pm 0.26$ | $\mathbf{10.63 \pm 0.66}$ | $11.73 \pm 0.54$ |

Figure 5: Training curves for IC3Net (top) and CommNet / DGN (bottom) on Box Pushing (left), Drone Scatter with stochastic evaluation (middle), and Drone Scatter with greedy evaluation (right)

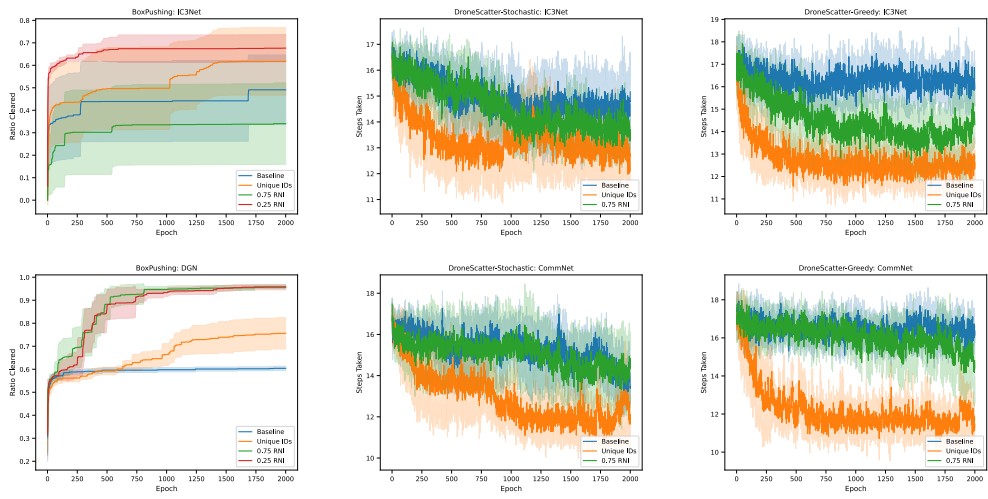

designed to separately test expressivity and symmetry-breaking. Ultimately, we find that, whilst unique IDs or smaller RNI augmentations can typically be applied without detriment on standard environments, they do not readily provide improved performance either. However, on environments where more complex coordination is required, these augmentations are essential for strong performance. With RNI and unique IDs being best suited to environments requiring increased expressivity and symmetry-breaking, respectively, it is interesting to note that no single method emerges which we can recommend as the *de facto* choice for MARL practitioners. This suggests that a more complete picture of the relationship between communication expressivity and downstream performance on relevant tasks remains an open question for future research. Furthermore, insights into GNN architectures can be leveraged in GDNs, which opens many promising avenues for future work in MARL communication.

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
