# OpenReview forum: "Universally Expressive Communication in Multi-Agent Reinforcement Learning"
_NeurIPS.cc/2022/Conference — NeurIPS 2022 Accept_

### Official Review · Reviewer_zdMz · 2022-07-07

**Rating:** 6
**Confidence:** 4
**Soundness:** 4 excellent
**Presentation:** 2 fair
**Contribution:** 2 fair

**Summary:**

This work studies the expressivity of GNN when used as a communication function within multi-agent reinforcement learning systems (MARL) and defines a formal framework called graph decision networks (GDN) and observation extension procedures to improve expressivity.

**Questions:**

Please read above

* In section 2 background the MARL introduction assumes a lot of previous knowledge from the reader, I understand that there is limited space, but authors could elaborate a bit on parameter sharing.


I believe this is specially important for stabilising a "Universally Expressive" framework. As the authors point out they are not the first to use parameter sharing with the end of stabilise training, but it is an assumption that is not present on most of SOTA MARL algorithms, e.g. MAPPO, IPPO, QMIX, MAVEN....



**Limitations:**

Authors do set clearly the limited scope of their review and analysis, although I believe that having parameter sharing in all the empirical evaluation gives an additional limitation of the analysis that could have been avoided,

**Strengths And Weaknesses:**

Authors provide a sound contribution. First, theoretically with their analysis of communication expressiveness in exiting techniques while introducing their framework. second with an extensive empirical evaluation with multiple baselines and environments.

Regarding originality, I believe the paper may be labelled as incremental since the framework and analysis are about augmenting expressivity of existing baselines.

About clarity, due to the theoretical nature of the paper it is hard to read. In this regard I believe that brief paragraphs explaining the directions of each section would greatly improve readability. Also at first read is not clear what are the significances of the GDN and what is analysing existing work, some check on that would help.

Last, I believe that the significance of the work is low, and this is my biggest concern with this work. It is concerned about communication in MARL but only with GNNs, I believe authors could do more to highlight the relevance of this study. There are a plethora of works about MARL communication and emergent languages which are not concerned about GNNs. Given the large scope of this conference I believe some comparison with those should be included.


Given all of this, since I believe that evaluating significance is inevitably a subjective topic, I lean towards acceptance despite having my reservations towards novelty and significance.

---

> ### Author Response · Authors · 2022-08-01
> **Response to Reviewer zdMz**
>
> We thank the reviewer for a thorough and insightful review, and for the effort that went into it: we really appreciate you engaging with our work. Please see our responses to your queries below.
>
> **Strengths and Weaknesses 3:**
>
> We greatly appreciate your feedback on this. We will make efforts to make the theoretical aspects of the paper more understandable from the perspective of MARL practitioners, given that the paper relies heavily on also having a background in GNNs.
>
> **Strengths and Weaknesses 4:**
>
> We agree that our work only applies to MARL communication that uses GNN-style message passing communication. However, there is a significant trend within MARL communication research towards such models (https://arxiv.org/pdf/2203.08975.pdf). Furthermore, as we note on lines 28 and 29, not all of these models are stated explicitly in terms of GNNs, but they can be captured within the framework of GDNs (for example: CommNet).
>
> It is true that some MARL communication models do not fall within the GDN paradigm, e.g. RIAL, DIAL, and ETCNet use a fixed message-passing structure, ATOC and BiCNet use an LSTM for combining messages, and SchedNet concatenates messages. We will add information to our background about some of these other methods and why they do not fall within the GDN paradigm, just after line 29.
>
> With respect to research about emergent languages: we find such work to have fundamentally different aims to the models we are augmenting, even though it is very interesting. The sub-field aims to study how language emerges: the models are typically applied to simplistic examples to study whether shared languages can be developed (https://arxiv.org/pdf/2006.02419.pdf). Many of the environments do not even have multiple time steps or a dynamic environment. On the other hand, the models we are augmenting are frequently applied to state of the art MARL environments, with the aim of providing communication that enables agents to solve the tasks.
>
> **Questions:**
>
> Thank you for this feedback: we will aim to clarify better in the paper what we mean by parameter sharing.
>
> Most state of the art MARL algorithms do use, or at least allow for parameter sharing between agents. To provide a correction to the reviewer: parameter sharing is in fact used in MAPPO / IPPO (https://arxiv.org/pdf/2103.01955.pdf section 3), QMIX (https://arxiv.org/pdf/1803.11485.pdf appendix B.1 and appendix C.2), and MAVEN (https://arxiv.org/pdf/1910.07483.pdf appendix C.1).
>
> **Limitations:**
>
> We required parameter sharing to be able to use our theoretical analysis, and our empirical results were a demonstration of that theory in practice: as such, we used shared weights for all evaluations. However, using shared weights is already very common and well established in MARL.
>
> We appreciate the reviewer’s suggestion though, and do acknowledge that considering the case of non-shared weights for agent networks and GNNs which allow for communication between heterogeneous agents is very interesting; in future work, we are aiming to consider heterogeneous GNNs and agent networks that do not have shared parameters.

---

### Official Review · Reviewer_Kpp5 · 2022-07-10

**Rating:** 6
**Confidence:** 4
**Soundness:** 3 good
**Presentation:** 3 good
**Contribution:** 3 good

**Summary:**

The paper proposes to represent the problem of learning communication for multi-agent systems RL using Graph neural network representations.  In addition to the formalization, the paper explores the use of 2 techniques from the GNN literature: random noise, and unique node IDs, towards better expressivity in MARL.

**Questions:**

- You propose 3 desideratas for MARL communication (l.139-141):  In what way do the 2 proposed techniques (random noise and unique IDs) help achieve each of these?
- Much of the motivation seems to be focused on expressivity of the graph (with the goal of “universal expressivity”).  However I would think it’s as important (or more) for that level of expressivity to be efficiently learnable (i.e. learnable in a relatively small number of samples).  Can you explain your view on efficient learnability in the context of this work?  And what are the properties of the proposed methods in this dimension?  You point out (l.184) that many samples may be needed for unique node IDs; is this the case in your results?
- Your results are based on “value of a metric for a run to be the best value achieved during training” (l.246).  While this may be standard in the literature, it seems to obfuscate the question of learning efficiency and stability, which I view as important properties.  Can you include in your results for each technique what was the #training samples at which this best value was reached?   And total #training samples that you ran, in the case that this is not 5000 (as per l.243).  This would help get more nuanced understanding of learning efficiency, and also support some of your claimes, such as “RNI methods typically take longer to converge than the baseline and unique IDs” (l.269-270), and “unique IDs tend to yield less stable solutions” (l.286-287).


**Limitations:**

- The authors provide reasonable discussion of the technique limitations of their work, but I would like more detail on the empirical results, as per questions above.
- The authors do not raise ethical or social aspects of the work.  I don’t see any major ethical concerns related to this work.


**Strengths And Weaknesses:**

(+) The idea of exploring the GNN paradigm for learning communication protocols for MARL is intriguing, and brings some interesting new perspective on the problem.  There is some interest in evaluating the effectiveness of techniques such as random noise and unique IDs in the MARL setting.
(+) The paper has good coverage of related work.
(+) The paper includes a good number of baselines and domains in the empirical results.

(-) In terms of clarity the paper is at times hard for a reader with more RL expertise, but less GNN expertise, to fully understand. E.g. 1-WL graph colouring, which appears to be a key concept, is defined very briefly.  What is \alpha in your definition of graph (l.84)?
(-) The specific techniques used, namely random noise and unique node IDs, are not particularly insightful.  They seem to bring expressivity, but perhaps at the expense of learnability (more on this below).

---

> ### Author Response · Authors · 2022-08-01
> **Response to Reviewer Kpp5**
>
> We thank the reviewer for a thorough and insightful review, and for the effort that went into it: we really appreciate you engaging with our work. Please see our responses to your queries below.
>
> **Weakness 1:**
>
> We appreciate that the main body of the paper is very light on background information due to the space constraints. We provide a full background in the appendix, which is very thorough and introduces topics for both MARL and GNN practitioners. However, given that the main audience of this paper is MARL researchers, we will attempt to include more background information on GNNs in the main body of the paper.
>
> To clarify “what \alpha is in our definition of the graph on l.84”: this ‘a’ refers to the attribute function of the graph, which is defined in the appendix when we introduce attributed graphs. However, we recognize that this can be confusing without the information provided in the appendix, so we will remove it from the definition on l.84. Thank you for bringing this to our attention.
>
> **Question 1:**
>
> Theorems 2-5 prove that these properties hold and the computational efficiency of the augmentations is stated in lines 141-142. Furthermore, we explicitly state that these properties are satisfied on lines 173-174 and lines 195-196, using more intuitive language than the theorems. Our experimental results demonstrate that these augmentations provide the 3 properties in practice as well.
>
> However, we acknowledge that these may not be as accessible to MARL readers without a background in GNNs, so we will signpost to the upcoming theorems more clearly. Specifically, we will add a further sentence after line 142 explaining that, in the following sections, we provide theorems which prove that the properties are satisfied by the augmentations.
>
> **Question 2:**
>
> We have no theoretical results about learning efficiency, only empirical ones. Due to space constraints and our extensive evaluations, we could not present our learning curves in the main body of the paper, so they are provided in the appendix. Our empirical results show that the augmentations mostly converge at a similar rate, albeit sometimes a bit slower. There are cases in which the augmentations even end up converging faster than the baseline, for example: TARMAC-IC3Net on Predator-Prey.
>
> The properties of our methods in this dimension are stated in the main section of the paper on line 274 and fully shown by the learning curves in the appendix. The answer to the subsequent question may also be found by analyzing the result curves in the appendix: convergence speeds are typically comparable between unique IDs and the baseline. However, we appreciate that this information would be nice to see in the main section of the paper.
>
> As such, we have added an extra line to each table in the paper, specifying, for each method on each environment, what the average number of epochs is until the best performing model is found. We have included this change specifically in the uploaded revision, so that we may hear feedback on how it is presented and whether it sufficiently answers the questions raised.
>
> **Question 3:**
>
> We agree that it’s useful to know the number of training samples at which the best values were reached. However, we believe that this information is best represented in the learning curves provided in the appendix.
>
> To provide clarification about the total number of training samples: each epoch in one of our models consists of training for 5000 episodes, then evaluating for 100 episodes. There are 2000 epochs in every experiment run.
>
> Finally, with respect to the claims we make on lines 269-270 and lines 286-287: these are supported by the result curves provided in the appendix.

---

> ### Comment · Reviewer_Kpp5 · 2022-08-08
> **Post-response comment**
>
> I thank the authors for their detailed response to my questions and the concerns I raised. It seems many of the answers are provided in the appendix.  In light of this, I will raise my score to Weak Accept.  The main thing holding me back from a higher score is that I find the specific augmentations considered (Unique ID and RNI) not particularly insightful.

---

### Official Review · Reviewer_yud9 · 2022-07-11

**Rating:** 6
**Confidence:** 3
**Soundness:** 3 good
**Presentation:** 3 good
**Contribution:** 2 fair

**Summary:**

This work tackles the challenge of enabling GNNs to learn richer communication protocols for MARL. To this end, the authors initially formalise previous GNN-based approaches to learn communication protocols in MARL as a node prediction problem. The authors subsequently propose RNI and CLIP, two input preprocessing methods that improve the functional expressivity of previous GNN-based MARL approaches. The authors provide theoretical analysis to prove that combining GNNs with RNI/CLIP ensures the existence of GNNs capable of learning communication protocols that require (i) symmetry breaking and (ii) functional expressiveness beyond 1-WL. Empirical evaluation of the returns from combining previous GNN-based MARL approaches and RNI/CLIP shows the proposed approach produces improved performance in environments requiring symmetry breaking and functional expressiveness beyond 1-WL.

**Questions:**

**Questions**

1. Are there any interesting insights that can be obtained from the messages passed by agents equipped with RNI/CLIP (particularly in terms of symmetry-breaking)?
2. What are the possible solutions to improve RNI's slow convergence in environments not requiring symmetry breaking?

**Suggestions**
1. Despite the theoretical analysis on RNI/CLIP being useful, I believe most of the space allocated to this is better served for (i) demonstrating the convergence/learning progress of RNI/CLIP and (ii) showing RNI/CLIP learns useful messages for symmetry breaking or going beyond 1-WL expressivity. After all, these theoretical guarantees are mostly limited to only guaranteeing the existence of GNNs that can approximate the optimal communication protocol.

2. Although helping strengthen the claims made in this work, the wide-range of evaluated communication learning methods can be reduced to only include fewer algorithms to make the presented results more compact. Additional comparisons of RNI/CLIPs performance for other algorithms may subsequently appear in the Appendix.

**Limitations:**

The authors have adequately highlighted the limitations of their work through their experiments and the resulting analysis. In terms of societal impact, I do not think that this work requires additional information regarding its negative societal impact since its application is still rather limited to simple MARL environments.

**Strengths And Weaknesses:**

**Originality**

**1. (Minor weakness) The proposed approach is incremental relative to previous works.**

Despite the novelty of the proposed approaches' application to learning communication protocols in MARL, these techniques have previously been proposed for other prediction problems, as alluded to by the authors in Lines 144 and 175. Line 155 also indicates that some theoretical contributions are minor extensions of theorems proven in previous work. Nevertheless, this weakness is outweighed by the remaining theoretical and empirical studies that the authors have conducted to show that the proposed approach can learn communication protocols requiring symmetry breaking and going beyond 1-WL expressivity in MARL.

**Quality**

**1. (Major Strength) Result reproducibility.**

The authors have provided adequate descriptions of the model architecture and hyperparameters, environments, and experiment protocols that ensure their results are reproducible.

**2. (Major Strength) The experiment design adequately demonstrates the claims made in by the authors.**

The environment and baseline selection empirically highlight the claims of this work regarding RNI/CLIP's ability to learn improved communication protocols in environments requiring (i) symmetry-breaking or (ii) function estimators beyond the 1-WL expressivity. Note that the evaluation in Predator-Prey and Easy Traffic Junction also highlights the potential weaknesses of the proposed approach when applied to environments not requiring (i) or (ii). This provides valuable insights that can inform the readers when applying the proposed technique to other environments.

**3. (Minor weakness) Missing theoretical/empirical analysis to strengthen the claims of this work further.**

Despite ensuring the existence of a learned GNN that can help with (i) symmetry-breaking, I believe the theoretical analysis on the ability of RL-based optimisation techniques to learn such GNNs is lacking. Nonetheless, the empirical demonstration of RNI/CLIP's performance indicates that this is not a significant issue for the evaluated environments. Furthermore, since the work mentioned convergence as a potential issue with a few proposed approaches, it is crucial to report the learning curve from the experiments in the experiment section (although this is provided in the Appendix).

That aside, it will also be interesting to analyze messages that RNI/CLIP passes in environments requiring (i) or (ii). Specifically, in terms of proving the usefulness of RNI/CLIP in addressing (i), inspecting the passed representation may highlight how they aid the symmetry-breaking process.

**Clarity**

**1. (Strength) The paper is well-written.**

The document is generally well-written. The method description and related theoretical analysis were clear and concise. Furthermore, the experiment section clearly states the intention behind the design/selection of various baseline/environments for demonstrating the central claims of the work.

**Significance**

**1. (Strength) The work provides significant results for people working in applying GNNs for learning communication protocols for MARL.**

Despite the proposed approach mainly being incremental compared to previous work, the theoretical and empirical analysis provides findings useful for people working in communication for MARL. The way the authors highlighted the potential limitations of the work is specifically helpful for future research.

---

> ### Author Response · Authors · 2022-08-01
> **Response to Reviewer yud9**
>
> We thank the reviewer for a thorough and insightful review, and for the effort that went into it: we really appreciate you engaging with our work. Please see our responses to your queries below.
>
> **Quality 3:**
>
> We agree that theory on the ability of RL-based optimisation techniques to learn such GNNs would be an amazing achievement. However, existing literature on convergence guarantees for RL algorithms is already limited and very complex, and GNN papers on models with universal expressivity similarly do not provide theory on the ability of training paradigms to converge to the models they prove exist. We are grateful to the reviewer for pointing out that our empirical results demonstrate the ability of our proposed augmentations to learn the GNNs which we prove to exist theoretically.
>
> Furthermore, as you request, we would have liked to be able to provide the learning curves in the main section of the paper. However, due to how extensive our evaluations are and how much space the curves take up, we are forced to provide them in the appendix instead and refer the reader to them on line 207, to respect the space constraints of the paper.
>
> **Question 1:**
>
> We appreciate the suggestion to analyze messages that RNI/CLIP passes to highlight how they aid the symmetry-breaking process. For example, one could measure and report the diversity in the set of messages, expecting higher diversity when the policy has better learned to perform symmetry-breaking. However, this is something that we feel is best left for future work.
>
> **Question 2:**
>
> Our experiments showed that using lower amounts of RNI resulted in faster and more reliable convergence. To further improve the slow convergence, we could try even lower amounts of RNI, since the universality results hold for any amount of RNI. However, the results of Abboud et al. [1] show that when using only a single RNI value, the performance is poor, so there is a point at which the amount of RNI can become too low. Abboud et al. [1] found good performance with RNI values ranging from 12.5% to 87.5%, so lower values of RNI could be promising to check in future evaluations. In our evaluations, we could only use two values from this range due to our limited computational budget.
>
> Furthermore, convergence speed depends greatly on the baseline being augmented. For example: IC3Net consistently converges quickly and reliably. Thus, augmenting a good baseline is also a promising way to ensure that RNI methods converge well.
>
> **Suggestion 1:**
>
> We appreciate the reviewer’s suggestion here, but we find the theoretical analysis to be crucial to present in the main body of the paper; it establishes how the augmentations are able to solve problems that the baselines are proven to be unable to solve, and provides principled reasons for using the augmentations in practice.

---

> > ### Comment · Reviewer_yud9 · 2022-08-08
> > **Response to Author Rebuttal**
> >
> > Thank you for providing answers to the questions that have been raised in the reviews. I am satisfied with most of the authors' answers.
> >
> > I still have the same concerns as reviewer Kpp5 regarding the lack of empirical results provided in the main text. While the authors argued for the importance of current theoretical results regarding the expressivity of GNNs for universally expressive communication, I believe that theoretical/empirical results on learning such GNNs via RL-based optimisation is a more impactful contribution to the research community (i.e. it not only demonstrates that a GNN exists to approximate any communication protocol, but also shows that such GNNs can be discovered via RL).
> >
> > Nevertheless, as suggested in my original review score, I still maintain a positive view about this work.

---

> > > ### Author Response · Authors · 2022-08-08
> > > **Response to empirical results being in appendices**
> > >
> > > We appreciate your feedback on this. There are two questions which we would like your further feedback on:
> > > 1. What are your thoughts on the extra information we provide in the tables of the revision we uploaded, and do these address your concerns about having further results in the main section of the paper?
> > > 2. If you were to select some of the training curves from the appendix to show in the main section of the paper, which ones would they be? We cannot include all of them, since they collectively take up 9 pages

---

> > > > ### Comment · Reviewer_yud9 · 2022-08-09
> > > > **Comments on Tables and Additional Training Curves**
> > > >
> > > > 1. If the purpose of including epochs was to illustrate the convergence rates of different experiment setups, I still believe that training curve figures is better than adding additional rows indicating the best performing epoch. Best performing epoch information can be rather deceptive in case evaluations in previous epochs produce slightly worse performance (i.e. not significantly worse based on the evaluated metric's confidence interval) before the best performing epoch. Nonetheless, it is still better than not including this information at all.
> > > >
> > > > 2. In general, it seems that the experiments conducted here varies across (a) MARL algorithm, (b) environment, and (c) observation augmentation method. While the in-depth analysis of the proposed methods across (a) is definitely one of the strengths of this paper, the authors can save space in the main text by only reporting the results in 2-3 MARL algorithms, reporting the results for the remaining algorithms in the appendix, and referring to these additional results in the main text. I do not have a specific preference over the reported algorithms as long as it is not MAGIC (i.e. because the main text mentioned that the results with this algorithm can be unstable). I leave the remaining selection over the algorithms to the authors.
> > > >
> > > > In terms of (b), you can limit the number of training curve images to 6-9 images (assuming results from 2-3 algorithms are reported) if only results from three environments are reported. These three environments will ideally be one of the benchmark environments (choose one), drone scatter, and box pushing.
> > > >
> > > > Overall, displaying the selected figures will take approximately 1 page. Also note that including some of the training curve figures will allow you to remove the tables from the main paper (i.e. information provided in the table can be inferred from the figures).

---

### Official Review · Reviewer_89rL · 2022-07-22

**Rating:** 6
**Confidence:** 4
**Soundness:** 3 good
**Presentation:** 3 good
**Contribution:** 3 good

**Summary:**

This paper attempts to improve communication in multi-agent systems by drawing from graph neural network techniques—they consider agents as nodes in a graph that allows techniques such as node labeling to apply in this setting. They introduce two augmentations to GNNs i.e., adding randomisation into node initialisation and a unique identifier method to differentiate agent ids in their observations,, and show how these algorithms allow more expressive communication. They evaluate this on standard benchmarks (3 different environments) and show that this improves performance in instances of symmetry breaking, but results in policies that are less than optimal in others.


**Questions:**

1. In line 23: why should the order of messages not matter? Even in some of the newer symbolic emergent communication settings, the ordering of distinct tokens matters in terms of comprehension of the message, and especially when moving to natural language, the history of messages/ordering of context makes a large difference to the meaning of the input. I understand that this paper does not deal with natural language, but it would be good to add to line 23 to explain why ordering should not matter, to clarify the distinction between permutation invariance in this setting, as opposed to other settings with richer language, where this should not hold.
2. Do you have insights on when the RNI vs. Unique ID should/shouldn’t yield improvements for different models/environments? Based on the results tables there seems to be clear difference in improvements of these two (they seem to work / not work symmetrically) and it would be helpful to try to pick those results apart to understand why this happens, to give us insight into the underlying working of these augmentations to the GNNs.


**Limitations:**

It would be good to have a larger discussion on when these GNN (with augmentation) methods should not be expected to yield better performance, since that’s helpful to our understanding of these newer methods. Overall I like this paper a lot, I think I would just like to understand it more!


**Strengths And Weaknesses:**

Strengths:
1. This paper is well written and clearly explained—nearly every question I had was answered immediately and was easy to follow.
2. They introduce a nice analogy between MARL communication methods and graph neural networks, where agents form the nodes in the graph (labeled with agent observations) and communication occur along the edges.
3. For the two augmentations to the GNNS that they propose, they clearly state out the procedure as well as prove why this should work.
4. They evaluate these augmentations empirically, in comparison to a baselines on several different standard MARL environments and show that both augmentations yield improved performance over baselines, for different types of models that they augment.

Weaknesses:

1. I like that there are definite gains in performance on adding the two augmentations over baselines, but I’m wondering if the gains might only be prevalent in the simplistic MARL environments used here? Do the authors have insights on the complexities of the different environments and whether some augmentations work better/worse for the different types of environments?
2. E.g., specifically for the unique node identifier augmentation, I’m curious to hear the authors thoughts on how this might impact performance if the agents are more diverse in terms of the attributes/skills/inventory items they possess (e.g., see environments like craftworld), and one unique identifier might not be enough information to be useful?

---

> ### Author Response · Authors · 2022-08-01
> **Response to Reviewer 89rL**
>
> We thank the reviewer for a thorough and insightful review, and for the effort that went into it: we really appreciate you engaging with our work. Please see our responses to your queries below.
>
> **Weakness 2:**
>
> No, a single unique identifier will always be sufficient to perfectly distinguish the agents, yielding universal expressivity and symmetry breaking. Our theoretical guarantees still hold in environments with more diverse agents, provided that the agents use shared weights. Diverse agents can be represented in a shared-weight setting by allocating a portion of their observations to specify their attributes / skills, like we state on lines 102-104 (arbitrary unique IDs will be concatenated in addition to this, one for each agent). Having a unique ID for each agent will still perfectly distinguish all of them from one another, and we see no reason why this would change performance in practice.
>
> As an aside, if we are considering the same craftworld as the reviewer, it appears not to be a multi-agent environment (https://arxiv.org/pdf/2011.00517.pdf).
>
> **Question 1:**
>
> We first note that there is a distinction that has to be made between the order in which messages are received from multiple agents, and the order in which tokens appear within a message. The order of words matters for natural language, and analogously in our case, the order of numbers in message vectors matters during message passing. However, when we state that “an agent’s policy should often not depend on the order in which messages are received at a given time step”, we are referring to the order in which messages from different agents are processed.
>
> There are some environments where an existing order is present, but we consider cases where it is not (i.e. parallel multi-agent environments). In our scenarios, any order imposed on the messages would be completely arbitrary. We prefer respecting the natural permutation invariance in our architectures, in alignment with much of the established work in MARL communication.
>
> **Question 2 / Weakness 1:**
>
> We do: our hypothesis for the superior performance of unique IDs when it comes to symmetry breaking is stated in lines 296 and 297 of the paper. With respect to why RNI performs better when it comes to expressivity, we postulate that it’s much easier for agents to overfit on the particular unique IDs given to them, since they are deterministically assigned. On the other hand, using RNI encourages the agents to learn policies which respect the permutation invariance between agents, since agents will receive different random observation augmentations at each time step.
>
> We will add the latter hypothesis into our paper, just after line 287.
>
> **Limitations:**
>
> The theory we present suggests that improved performance should be expected in scenarios where expressivity beyond 1-WL or symmetry breaking is required, and that performance should typically not degrade in scenarios where they are not. Our empirical results demonstrate this as well.
>
> We agree with the reviewer that it would be really interesting to see further results on other environments / scenarios, and we hope that this paper will serve as a starting point for others to incorporate these methods into their research.

---

### Author Response · Authors · 2022-08-01
**Aimed-for changes after initial reviews**

Based on the feedback received from all reviewers so far, we aim to make the following changes:

**Hypothesis for Superior Performance of RNI for Expressivity:**

Our hypothesis for the superior performance of unique IDs when it comes to symmetry breaking is stated in lines 296 and 297 of the paper. With respect to why RNI performs better when it comes to expressivity, we postulate that it’s much easier for agents to overfit on the particular unique IDs given to them, since they are deterministically assigned. On the other hand, using RNI encourages the agents to learn policies which respect the permutation invariance between agents, since agents will receive different random observation augmentations at each time step.

We will add the latter hypothesis into our paper, just after line 287.

**Remove ‘a’ from Graph Definition:**

​​To clarify what ‘a’ is in our definition of the graph on line 84: this ‘a’ refers to the attribute function of the graph, which is defined in the appendix when we introduce attributed graphs. However, we recognize that this can be confusing without the information provided in the appendix, so we will remove it from the definition on line 84.

**Segway into Theorems:**

We will add a further sentence after line 142 explaining that, in the following sections, we will provide theorems which prove that the 3 properties are satisfied by the augmentations.

**Parameter Sharing Clarification:**

We will aim to clarify better in the paper what we mean by parameter sharing.

**Extra Information in Tables (included in revision):**

We have added an extra line to each table in the paper, specifying, for each method on each environment, what the average number of epochs is until the best performing model is found. We have included this change specifically in the uploaded revision, so that we may hear feedback on how it is presented and whether it sufficiently answers the questions raised.

**Background on non-GNN-based Communication Models:**

It is true that some MARL communication models do not fall within the GDN paradigm, e.g. RIAL, DIAL, and ETCNet use a fixed message-passing structure, ATOC and BiCNet use an LSTM for combining messages, and SchedNet concatenates messages. We will add information to our background about some of these other methods and why they do not fall within the GDN paradigm, just after line 29.

---

### Meta-Review · Area_Chair_Vn9V · 2022-08-26

**Recommendation:** Accept
**Confidence:** Less certain

**Metareview:**

Reviewers found the paper's connections between MARL and GNNs interesting and well-written, and the experiments convincing. Given the unanimous support, I recommend acceptance. That said, I encourage the authors to integrate reviewer feedback, including trying to move some of the details and plots requested to the main text.

**Award:**

No

---

### Decision · Program_Chairs · 2022-09-14

Accept